# First field evaluation of novel LDH- and HRP2-based rapid tests for *Plasmodium vivax* and *Plasmodium falciparum* malaria diagnosis

Colins O. Oduma[1], Birhanu Lulu[2], Yalemwork Ewnetu[3], Laurel A. Lown[4], Tolulope Adeyemi Kayode[4], Dawit Hawaria[2], Cristian Koepfli[4]*

**1** Uzima University, Department of Microbiology, Kisumu, Kenya, **2** Malaria and Other Vector-Borne Diseases Research Center, Malaria and Other Vector-Borne Diseases Research Center, Hawassa University, Hawassa, Ethiopia, **3** Department of Medical Biotechnology, University of Gondar, Gondar, Ethiopia, **4** Department of Biological Sciences and Eck Institute for Global Health, University of Notre Dame, Notre Dame, Indiana, United States of America

\* ckoepfli@nd.edu

## Abstract

### Background

Rapid diagnostic tests (RDTs) are crucial for malaria diagnosis. Where *Plasmodium falciparum* and *Plasmodium vivax* are co-endemic, and where *P. falciparum hrp2/3* deletions are frequent, RDTs need to detect either species, and *P. falciparum* using additional antigens to HRP2, such as LDH.

### Methods

Clinical patients presenting for malaria diagnosis in southern Ethiopia were enrolled and tested by microscopy at the health center and by four different RDTs: (i) BIO-CREDIT Malaria Ag Pf (cHL) with a line combining HRP2 and LDH for *P. falciparum*, (ii) BIOCREDIT Malaria Ag Pf/Pv (cHL/L) with one line combining HRP2 and LDH for *P. falciparum* and one with LDH for *P. vivax*, (iii) Bioline Malaria Ag Pf/Pf/Pv with separate lines for HRP2 and LDH for *P. falciparum*, and LDH for *P. vivax*, and (iv) First Response with an HRP2 line for *P. falciparum* and a LDH line for *P. vivax*. The two BIOCREDIT RDTs had not previously been tested in the field. qPCR and expert microscopy were conducted as reference tests. *P. falciparum* positive samples were typed for *hrp2/3* deletion.

### Results

Among 708 patients included in the final analysis, 46.0% were positive by qPCR (77 *P. falciparum* mono-infections, 198 *P. vivax* mono-infections, and 51 mixed infections). Strong agreement was observed between results of the different RDTs, with no significant differences in sensitivity. At densities >20 parasites/µL by qPCR, all

**Data availability statement:** All data are in the manuscript and the supporting information files (S2 File).

**Funding:** This study was funded by Rapigen Inc. through a grant to the University of Notre Dane for CK to conduct this study (grant number 24-0341). The authors have discussed the results with Rapigen prior to publication, and jointly decided to publish. It was the responsibility of the corresponding author (CK) to ensure all data is accurate, and that data interpretation is correct. Beyond the finding provided for this study, Rapigen did not make any payments to any of the authors. None of the authors holds any intellectual property rights in the BIOCREDIT rapid tests, or shares of Rapigen.

**Competing interests:** The authors have declared that no competing interests exist.

RDTs reached sensitivities of >96% for *P. falciparum*, compared to 63% by health center microscopy, and for *P. vivax* all RDTs reached sensitivities of >92%, compared to 72% by health center microscopy. Specificity was >99% for all *P. falciparum* RDTs and >98% for all *P. vivax* RDTs. Only 2/53 *P. falciparum* infections typed carried *hrp2* and *hrp3* deletions, both were detected by all LDH-based RDTs.

## Conclusions

Use of RDTs improves diagnostic accuracy compared to microscopy. The novel BIO-CREDIT and Bioline RDTs show high sensitivity and specificity for *P. falciparum* and *P. vivax* diagnosis.

## Author summary

Rapid diagnostic tests (RDTs) are key for malaria control. In many countries in the Horn of Africa, Latin America, and the Asian Pacific, *P. falciparum* and *P. vivax* are co-endemic. Further, while the HRP2 protein is the most sensitive target for *P. falciparum* RDTs, it can be deleted from the parasite's genome, resulting in false-negative tests. Where this is the case, RDTs need to be able to detect alternative proteins, such as LDH. Some RDTs can detect *P. vivax*, and *P. falciparum* HRP2 and LDH though three separate test lines. New RDTs combine *P. falciparum* HRP2 and LDH into a single test line. These RDTs might reduce errors in test interpretation, as a single line indicates a *P. falciparum* infection, rather than three possible results (HRP2 only, LDH only, or HRP2 and LDH). In this study, RDTs with two vs. three lines performed virtually identical among clinical patients in Ethiopia. Compared to qPCR, at densities above 20 parasites per µL of blood, they detected >96% of *P. falciparum* and >92% of *P. vivax* infections. All RDTs detected substantially more infections than microscopy which is still widely used in health centers.

## Introduction

Effective and timely diagnosis is essential for patient management. Historically, light microscopy had been the main method for diagnosis. Expert microscopists detect infections at densities <100 parasite/µL [1]. Field microscopy, routinely conducted at health centers by microscopists that underwent basic training, is typically substantially less accurate [2, 3]. In the last two decades, rapid diagnostic tests (RDTs) became a key alternative tool for malaria diagnosis in health centers and for programs such as active or reactive case detection [4, 5]. RDTs are lateral flow devices that detect parasite antigens in the blood of infected individuals through immunohistochemistry. Their use requires minimal training, results are available within approximately 30 minutes, and the cost is less than USD1 per test. Over 300 million RDTs are procured each year by malaria control programs.

RDTs detect antigens such as Histidine Rich Protein (HRP2) that is exclusively expressed by *Plasmodium falciparum*, or Plasmodium Lactate Dehydrogenase (pLDH) that is expressed by all *Plasmodium* parasite species. For *P. falciparum*, HRP2-based RDTs exhibit the highest sensitivity. These RDTs also detect HRP3 due to cross-reactivity. Their use is threatened by deletions of the *hrp2* and *hrp3* genes which result in false-negative diagnosis. The WHO recommends to use alternative diagnostics if the prevalence of *hrp2* deletion is above 5% [6]. *hrp2/3* deletions are particularly frequent in South America, and in the Horn of Africa, e.g., Ethiopia [7, 8].

RDTs with lines for HRP2 and LDH allow detection of parasites even in the case of *hrp2/3* deletion. If the test lines are separate, they allow the identification of infections possibly harboring a deletion, *i.e.*, when the LDH line is positive but the HRP2 line is negative. Confirmation of deletions requires further typing through molecular methods. In health centers, interpretation of results of RDTs with two lines for *P. falciparum* may be more error prone, as either one or two positive lines indicate an infection. As an alternative, the HRP2 and LDH-targets can be combined into a single line. This line will show a positive result if LDH and/or HRP2 is detected, but will not allow for identifying infections with possible deletion. In regions where multiple *Plasmodium* species are endemic, RDTs ideally contain additional lines detecting either all species (pan-specific lines), or single species, e.g., *P. vivax*.

In Ethiopia, *P. falciparum* and *P. vivax* are co-endemic, and the frequency of *hrp2* and *hrp3* deletion is high [9–11]. In this study, we compared different RDTs for *P. falciparum* and/or *P. vivax* among clinical patients presenting to health centers, including novel RDTs detecting *P. falciparum* through a single line combining HRP2 and LDH. All RDTs evaluated were designed for the diagnosis of clinical *P. falciparum* or *P. vivax* infections. Two of the RDTs (BIOCREDIT Malaria Ag Pf (cHL) and BIOCREDIT Malaria Ag Pf/Pv (cHL/L)) had not previously been tested in the field. Few studies directly compared RDTs with separate HRP2 and LDH lines to those with combined lines [12].

## Methods

### Ethical approval

Informed written consent was collected from each adult, or, in the case of minors, from the legal guardian prior to sample collection. This study was approved by the Hawassa University College of Medicine and Health Sciences Institutional Review Board (ref. no. IRB/355/15), and the University of Notre Dame Institutional Review Board (approval no. 23-07-7990).

### Study design

A prospective, consecutive study among patients presenting for malaria diagnosis was conducted. Patients above 1 year of age were eligible to be enrolled at three health centers within Hawassa City in southern Ethiopia (Millennium, Tilte, and Alamura). Patients with signs of severe malaria were excluded. Sample collection was conducted between February 14 and March 19, 2024. 200–250 µL of capillary blood was collected in microtainer EDTA tubes by finger prick. Screening by RDT and local microscopy were performed on-site, a slide was prepared for expert microscopy, and the remaining blood was stored at -20°C until DNA extraction. Study participants were treated as per the national guidelines by healthcare providers at the health centers.

### Sample size considerations

We assumed that 20% of all individuals would be positive for *P. falciparum* at a density of >20 parasites/µL, and 20% of individuals would be positive for *P. vivax* at a density of >20 parasites/µL. We further assumed (based on our previous studies on the limit of detection of novel RDTs for *P. falciparum* [13, 14]), that at this density RDTs would detect at least 90% of infections. Based on these assumptions, aiming for a 95% confidence interval spanning not more than 5%, 691 samples are required [15]. We initially aimed for 750 samples, 708 were included in the final analysis.

## Malaria parasite diagnosis by rapid diagnostic tests

Four different malaria antigen RDT kits were compared: (i) the BIOCREDIT Malaria Ag Pf (cHL) (lot no. DTC026 and DTC037, expiry November 2025, 'cHL' stands for 'combined HRP2/LDH') manufactured by Rapigen, (ii) the BIOCRE-DIT Malaria Ag Pf/Pv (cHL/L) (lot no. DTC072 and DTC060, expiry November 2025) manufactured by Rapigen, (iii) the Bioline Malaria Ag Pf/Pf/Pv (lot no. 05GDJ001C, expiry June 2026), manufactured by Abbott Diagnostics, and (iv) the First Response Malaria Ag Pf/Pv (lot no. 76D0222S, expiry March 2024), manufactured by Premier Medical Corporation. The BIOCREDIT RDTs are considered index tests for this study and are currently undergoing WHO pre-qualification. The Bioline Malaria Ag Pf/Pf/Pv and First Response Malaria Ag Pf/Pv are considered reference tests and received WHO pre-qualification in 2018 [6].

The BIOCREDIT Malaria Ag Pf (cHL) RDT contains a single line combining HRP2 and pLDH for detection of *P. falciparum*. The BIOCREDIT Malaria Ag Pf/Pv (cHL/L) contains two lines. One line of HRP2 and pLDH combined for the detection of *P. falciparum*, and a second line of pLDH for detection of *P. vivax*. The First Response Malaria Ag Pf/Pv contains two lines, one of HRP2 for *P. falciparum,* and one of pLDH for *P. vivax*. The Bioline Malaria Ag Pf/Pf/Pv RDT contains three lines. A HRP2 line for the detection of *P. falciparum*, a second line of pLDH for the detection of *P. falciparum*, and third line of pLDH for the detection of *P vivax*. The First Response RDT was used by the health centers at the time of the study. Due the local availability of the RDT, 524/708 samples were screened using it.

Blood samples were tested by the two BIOCREDIT RDTs and the First Response RDT in the health center immediately upon sample collection (without blinding of study personnel to the results of the index or reference tests). Due to delays in importing the Bioline RDT, it was run on whole blood kept at -20°C in EDTA tubes for approximately 6 months. While freezing is not recommended by WHO guidelines for RDT testing and might impact sensitivity, a single freeze-thaw cycle as done for this study was shown not to affect HRP2 concentrations [16]. Test procedures were identical for all RDTs. Approximately 5 μL of blood was put onto the sample spot on the test kit, followed by 2–3 drops of assay diluent buffer on a separate spot. Results were read after approximately 20–30 minutes as per manufacturer instructions. Tests were considered invalid and repeated if the control line was not positive.

## Expert microscopy and qPCR

For expert microscopy, 3 μL of thick and 2 μL thin blood smears were prepared and stained according to WHO standards [7]. Slides were read by two Level 2 microscopists that were blinded to each other, and a Level 1 microscopist in case of discrepant results. All microscopists were blinded to RDT and qPCR results. Asexual parasite density was calculated whenever a slide was reported as positive. A slide was declared negative after examining 100 microscopic fields.

DNA extraction and qPCR were conducted using a portable laboratory setup described previously [17]. In brief, DNA was extracted from 100 μL blood using the Genomic DNA Extraction kit (Macherey-Nagel, Düren, Germany) and eluted in an equivalent volume of elution buffer. 4 μL of DNA, corresponding to 4 μL of blood was screened for *P. falciparum* and *P. vivax* separately on the MIC qPCR instrument (Bio Molecular Systems). *P. falciparum* was detected using the varATS assay which amplifies a target present in approximately 20 copies per parasite [18], and *P. vivax* was detected using the *cox1* assay [19], which amplifies a mitochondrial gene present in approximately 10 copies per parasite. The 95% limit of detection is approximately 0.3 parasites/μL blood for *P. falciparum*, and slightly higher for *P. vivax* [20]. Absolute parasite densities for *P. falciparum* and *P. vivax* were obtained using an external standard curve quantified by droplet digital PCR (ddPCR). In the case of *P. vivax*, a higher density blood sample identified by microscopy was used to construct the standard curve, and for *P. falciparum*, 3D7 cultured parasites were used. Individuals conducting PCR were not blinded to RDT results and results from health center microscopy. No qPCR screening for *P. malariae* or *P. ovale* was done.

*P. falciparum* infections at densities above 10 parasites/μL were typed for *hrp2* and *hrp3* deletions by digital PCR [21]. A published protocol was modified to multiplex assays for *hrp2*, *hrp3*, and a single-copy control gene (*serine-tRNA ligase*,

herein referred to as *tRNA*) in a single reaction and run on the Qiagen QIAcuity digital PCR instrument. Published primer and probe sequences were used [21], as sole modification the *hrp3* probe was coupled with Texas Red as reporter dye. The reaction mix contained: 3 µL QIAcuity Probe PCR Kit, 1.6 µM each of *hrp2* forward and reverse primer, 0.1 µM of *hrp2* probe, 0.8 µM each of *hrp3* forward and reverse primer, 0.05 µM of *hrp3* probe, 1.6 µM each of *tRNA* forward and reverse primer, 0.1 µM of *tRNA* probe, 2 µL DNA, and $H_2O$ for a total volume of 12 µL. Amplification conditions were as follows: 2 minutes at 95°C, followed by 50 cycles of 15 seconds at 95°C and 1 minute at 56°C. All protocols for qPCR and dPCR are provided in S1 File.

## Data analysis

All data is available in S2 File. Sensitivity was calculated as the proportion of infections detected by RDT or microscopy relative to infections detected by qPCR as gold standard, and against thresholds of all densities, 20 and 200 parasites/µL as determined by qPCR. Specificity was calculated as the proportion of qPCR-negative samples that were negative by RDT or microscopy. Differences in sensitivity between diagnostic methods were tested using Chi-square test or Fisher's exact test whenever appropriate. All reporting followed STARD guidelines for diagnostic accuracy studies [22] (S3 File).

PCR correction of expert microscopy results was conducted to account for potential mix up between *P. falciparum* and *P. vivax*. Corrections were applied as follows: (i) If expert microscopy identified *P. vivax*, but qPCR detected *P. falciparum* mono-infection, the result was corrected to *P. falciparum* (n = 4). (ii) If expert microscopy identified *P. falciparum*, but qPCR detected *P. vivax* mono-infection, the result was corrected to *P. vivax* (n = 1). (iii) If expert microscopy identified mixed infection with *P. falciparum* and *P. vivax*, but qPCR detected only *P. vivax*, the result was corrected to *P. vivax* (n = 13). (iv) No cases were observed in which expert microscopy identified mixed infection and qPCR detected only *P. falciparum*. (v) If expert microscopy identified *P. vivax* and/or *P. falciparum* but qPCR detected no parasites, the result was corrected to negative (n = 7). No corrections were made if qPCR detected a mixed infection, but only one species was detected by expert microscopy (n = 19).

The limit of detection (LoD) was defined as the lowest parasite density at which the probability of a positive RDT result reached 95%. The binary RDT outcome (positive/negative) was modeled as a function of log10-transformed parasite densities by qPCR, using a generalized linear model with a probit link, which models detection probability as the cumulative normal distribution of parasite density. The fitted model provided a detection probability curve, and the LoD was identified as the parasite density at which this curve crossed the 95% detection probability. Confidence intervals for the LoD were calculated by numerically solving for the parasite densities where the 95% confidence bounds of the fitted curve intersected the 95% detection probability.

## Results

A total of 734 individuals presenting for malaria diagnosis were enrolled. Blood was not available from 26 individuals to run the Bioline RDT, and thus were excluded from analysis. The final dataset consisted of 708 samples collected in three health centers (448 in Millenium, 141 in Tilte, and 119 in Alamura, Fig 1). All data is presented for the three health centers combined. The majority of the clinical patients (602/708, 85.0%) reported onset of clinical symptoms within 5 days before seeking health care. Most patients were adults above 15 years (585/708, 82.6%), followed by school age children of 5–15 years (102/708, 14.4%), while few young children under 5 years presented for diagnosis (21/708, 3.0%). The majority (54.5%) of patients were female. Only 23.2% (164/544) of study participants reported having slept under a bed net the night before sample collection.

Overall, 326/708 (46.0%) of individuals tested positive by qPCR; 77 (10.9%) carried *P. falciparum* mono-infections, 198 (28.0%) carried *P. vivax* mono-infections, and 51 (7.2%) carried mixed *P. falciparum* and *P. vivax* infections. Positivity by qPCR was moderately higher in males compared to females (162/322, 50.3% vs. 164/386, 42.5%, *P* = 0.045). Test

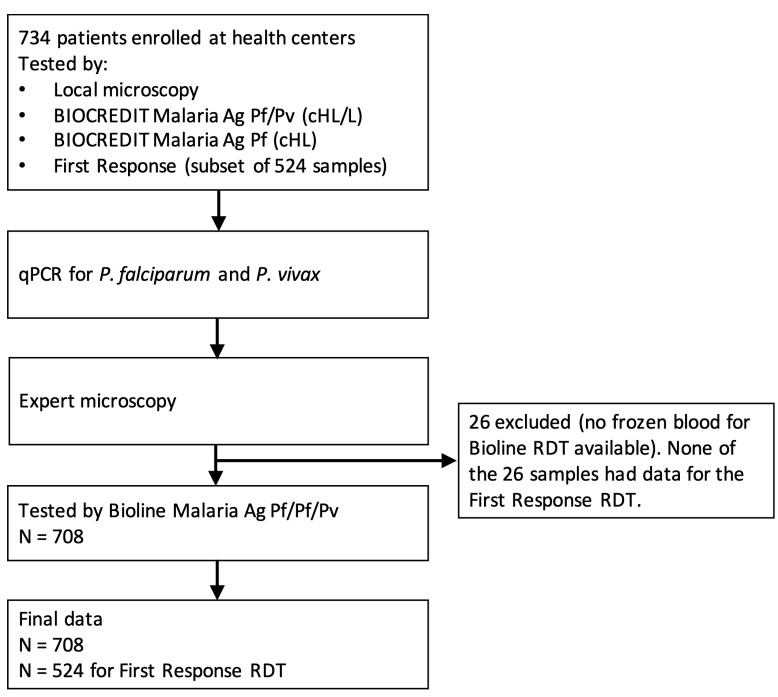

**Fig 1. Flow of samples.**

positivity rates did not differ significantly between young children aged <5 years (8/21, 38.1%), school children aged 5–15 years (49/102, 48.0%) and adults aged >15 years (269/585, 46.0%, $P = 0.705$).

By microscopy done at the health center, 139/708 (19.6%) samples were positive. 41 (5.8%) were *P. falciparum* mono-infections, 96 (13.6%) were *P. vivax* mono-infections, and 2 (0.3%) were mixed infections.

### *P. falciparum* diagnosis by RDT

Sensitivity and specificity of the two BIOCREDIT and the Bioline RDTs for *P. falciparum* are given in Table 1. Using qPCR as gold standard, and including infections of all densities, RDT sensitivity of the Bioline and Biocredit RDTs ranged from 46.9% to 48.4%. On the Bioline RDT, the HRP2 line detected 59/128 infections, while the LDH line detected 53/128 infections ($P = 0.450$). At densities >20 parasites/µL by qPCR, RDT sensitivity ranged from 88.3% to 96.7% with no significant differences among RDTs ($P = 0.251$), and at densities >200 parasites/µL from 94.0% to 100% ($P = 0.533$). All RDTs were more sensitive than local microscopy at the health center (all densities: $P < 0.001$, >20 parasites/µL $P < 0.001$, >200 parasites/µL $P < 0.001$ for all RDTs, and for the HRP2 and LDH line on the Bioline). Specificity was >99% for all RDTs (Table 1). Table 2 includes data for the reduced set of 524 samples also screened by the First Response RDT; results were near-identical for all four RDTs (Table 2). Using expert microscopy as reference, sensitivity of all RDTs ranged from 91% to 97% (Tables 1, 2).

The limit of detection (LOD), calculated as 95% probability that an infection is detected, was 33–64 parasites/µL for all RDTs with an HRP2 line, and 254 and 274 parasites/µL for the LDH line on the Bioline RDT for the two datasets, respectively (Table 1).

On the two BIOCREDIT RDTs, the combined HRP2/LDH line yielded the same result for 57/58 *P. falciparum*-positive samples. A single infection at a density of 20 parasites/µL by qPCR was detected by the Pf/Pv test only.

**Table 1. Sensitivity and specificity of RDTs and health center microscopy for *P. falciparum* using qPCR and expert microscopy as reference. N = 708.**

| Diagnostic method (N = 708) | 95% LOD [CI95] | Sensitivity vs qPCR | | | Sensitivity vs expert microscopy | Specificity vs qPCR |
|---|---|---|---|---|---|---|
| | | All densities (n/N) | > 20 parasites/µL (n/N) | >200 parasites/µL (n/N) | | |
| Health center microscopy | 15712.1 [5051.8, 48867.6] | 30.5% [22.6, 39.2] (39/128) | 63.3% [49.9, 75.4] (38/60) | 68.0% [53.3, 80.5] (34/50) | 71.1% [54.1, 84.6] (27/38) | 99.3% [98.2, 99.8] (576/580) |
| BIOCREDIT Malaria Ag Pf (cHL) | 64.4 [15.4, 268.8] | 47.7% [38.8, 56.7] (61/128) | 96.7% [88.5, 99.6] (58/60) | 98.0% [89.4, 99.9] (49/50) | 94.7% [82.3, 99.4] (36/38) | 99.7% [98.8, 100.0] (578/580) |
| BIOCREDIT Malaria Ag Pf/Pv (cHL/L) | 51 [12.0, 217.3] | 48.4% [39.5, 57.4] (62/128) | 96.7% [88.5, 99.6] (58/60) | 98.0% [89.4, 99.9] (49/50) | 94.7% [82.3, 99.4] (36/38) | 99.7% [98.8, 100.0] (578/580) |
| Bioline Malaria Ag Pf/Pf/Pv any line | 35.6 [0.9, 1346.4] | 46.9% [38.0, 55.9] (60/128) | 96.7% [88.5, 99.6] (58/60) | 100% [92.9, 100.0] (50/50) | 97.4% [86.2, 99.9] (37/38) | 99.4% [98.5, 99.9] (577/580) |
| Bioline Malaria Ag Pf/Pf/Pv LDH line | 254.2 [29.8, 2169.2] | 41.4% [32.8, 50.4] (53/128) | 88.3% [77.4, 95.2] (53/60) | 94.0% [83.5, 98.7] (47/50) | 92.1% [78.6, 98.3] (35/38) | 99.4% [98.5, 99.9] (577/580) |
| Bioline Malaria Ag Pf/Pf/Pv HRP2 line | 54.1 [3.1, 934.4] | 46.1% [37.2, 55.1] (59/128) | 95.0% [86.1, 99.0] (57/60) | 98.0% [89.4, 99.9] (49/50) | 97.4% [86.2, 99.9] (37/38) | 99.7% [98.8, 100.0] (578/580) |

**Table 2. Sensitivity and specificity of RDTs and health center microscopy for *P. falciparum* using qPCR and expert microscopy as reference. Reduced dataset of N = 524 samples with data for the First Response RDT available.**

| Diagnostic method (N = 524) | 95% LOD [CI95] | Sensitivity | | | Sensitivity vs. expert microscopy | Specificity |
|---|---|---|---|---|---|---|
| | | All densities | > 20 parasites/µL % (n/N) | >200 parasites/µL % (n/N) | | |
| Health center microscopy | 8598.3 [2416.2, 30598.0] | 36.0 [26.1, 46.8] (32/89) | 60.8 [46.1, 74.2] (31/51) | 65.1 [49.1, 79.0] (28/43) | 73.5% [55.6, 87.1] (25/34) | 99.1 [97.7, 99.7] (431/435) |
| BIOCREDIT Malaria Ag Pf (cHL) | 33.8 [3.2, 359.9] | 58.4 [47.5, 68.8] (52/89) | 98.0 [89.6, 100.0] (50/51) | 100.0 [91.8, 100.0] (43/43) | 97.1% [84.7, 99.9] (33/34) | 99.5 [98.3, 99.9] (433/435) |
| BIOCREDIT Malaria Ag Pf/Pv (cHL/L) | 33.8 [3.2, 359.9] | 58.4 [47.5, 68.8] (52/89) | 98.0 [89.6, 100.0] (50/51) | 100.0 [91.8, 100.0] (43/43) | 97.1% [84.7, 99.9] (33/34) | 99.5 [98.3, 99.9] (433/435) |
| Bioline Malaria Ag Pf/Pf/Pv any line | 35.6 [1.1, 1129.2] | 57.3 [46.4, 67.7] (51/89) | 96.1 [86.5, 99.5] (49/51) | 100.0 [91.8, 100.0] (43/43) | 97.1% [84.7, 99.9] (33/34) | 99.3 [98.0, 99.9] (432/435) |
| Bioline Malaria Ag Pf/Pf/Pv LDH line | 274.0 [30.5, 2463.4] | 50.6 [39.8, 61.3] (45/89) | 88.2 [76.1, 95.6] (45/51) | 93.0 [80.9, 98.5] (40/43) | 91.2% [76.3, 98.1] (31/34) | 99.3 [98.0, 99.9] (432/435) |
| Bioline Malaria Ag Pf/Pf/Pv HRP2 line | 58.7 [3.8, 906.4] | 56.2 [45.3, 66.7] (50/89) | 94.1 [83.8, 98.8] (48/51) | 97.7 [87.7, 99.9] (42/43) | 97.1% [84.7, 99.9] (33/34) | 99.5 [98.3, 99.9] (433/435) |
| First Response | 63.2 [10.7, 372.2] | 57.3 [46.4, 67.7] (51/89) | 96.1 [86.5, 99.5] (49/51) | 97.7 [87.7, 99.9] (42/43) | 94.1% [80.3, 99.3] (32/34) | 99.8 [98.7, 100.0] (434/435 |

53 *P. falciparum* infections were successfully typed for *hrp2/3* deletions. 45.3% (24/53) were wild type, 51.0% (27/53) carried *hrp3* deletions but no *hrp2* deletions, and 3.8% (2/53) carried *hrp2* and *hrp3* deletion. The two samples with double deletion all had densities >1000 parasites/µL and were detected by the BIOCREDIT and Bioline RDTs though their LDH targets. Of note, one was also detected by the First Response RDT, even though this RDT only has an HPR2 line for *P. falciparum*.

### *P. vivax* diagnosis by RDT

Sensitivity and specificity of RDTs for *P. vivax* using qPCR as gold standard are given in Table 3. Including all infections detected by qPCR, sensitivity ranged from 45.4% to 46.4%, with no significant differences between RDTs (*P* = 0.980). At >20 parasites/µL by qPCR, sensitivity ranged from 92.6% to 96.4% (*P* = 0.450), and at densities >200 parasites/µL from

**PLOS Neglected Tropical Diseases**

**Table 3. Sensitivity and specificity of RDTs and health center microscopy for *P. vivax* using qPCR and expert microscopy as reference. N = 708.**

| Diagnostic method (N = 708) | 95% LOD [CI95] | Sensitivity vs qPCR | | | Sensitivity vs. expert microscopy | Specificity vs qPCR |
|---|---|---|---|---|---|---|
| | | All densities % (n/N) | > 20 parasites/µL % (n/N) | >200 parasites/µL % (n/N) | | |
| Health center microscopy | 5118.3 [2597.7, 10084.9] | 36.5% [30.6, 42.9] (91/249) | 72.7% [63.9, 80.4] (88/121) | 80.0% [71.3, 87.0] (88/110) | 76.0% [66.6, 83.8] (79/104) | 98.5% [96.9, 99.4] (452/459) |
| BIOCREDIT Malaria Ag Pf/Pv (cHL/L) | 208.7 [53.1- 821.3] | 45.4% [39.1, 51.8] (113/249) | 92.6% [86.3, 96.5] (112/121) | 98.2% [93.6, 99.8] (108/110) | 92.3% [85.4, 96.6] (96/104) | 99.3% [98.1, 99.9] (456/459) |
| Bioline Malaria Ag Pf/Pf/Pv | 230 [85.4, 619.9] | 45.8% [39.5, 52.2] (114/249) | 92.6% [86.3, 96.5] (112/121) | 97.3% [92.2, 99.4] (107/110) | 91.3% [84.2, 96.0] (95/104) | 99.1% [97.8, 99.8] (455/459) |

97.3% to 98.7% (*P* = 0.772). RDTs were significantly more sensitive than local microscopy (all densities: *P* = 0.095, > 20 parasites/µL: *P* < 0.001, > 200 parasites/µL: *P* < 0.001). Specificity was > 98% for all RDTs. Data for the reduced dataset of 524 samples also screened by the First Response RDT is given in Table 4. Sensitivities were similar to the set of 708 samples, and specificity was above 98% in both sets of samples. Using expert microscopy as reference, sensitivity of all RDTs ranged from 91% to 97% in both sets of samples (Tables 3, 4).

The LOD was 208–230 parasites/µL for the BIOCREDIT and Bioline RDTs in the full dataset (Table 3), and was 85–180 parasites/µL for the three RDTs used to screen the 524 set of samples (Table 4).

### Detection of mixed-species, and very low-density infections

Among 12 samples with *P. falciparum* and *P. vivax* mixed-infection, each at >20 parasites/µL, both species were detected in only one sample by health center microscopy. In contrast, the BIOCREDIT RDT detected 5/12, and the Bioline RDT detected 7/12 mixed-species infections. By expert microscopy, five mixed-infections were detected.

At densities <20 parasites/µL, as expected, RDTs showed poor sensitivity. For *P. falciparum*, the BIOCREDIT Malaria Ag Pf (cHL) detected 4.4% (3/68), the BIOCREDIT Malaria Ag Pf/Pv (cHL/L) 5.9% (4/68), the Bioline Malaria Ag Pf/Pf/Pv 2.9% (2/68), and the First Response 5.3% (2/38) of infections. For *P. vivax*, the BIOCREDIT Malaria Ag Pf/Pv (cHL/L) detected 1.1% (1/128), the Bioline Malaria Ag Pf/Pf/Pv 1.6% (2/128), and the First Response 1.1% (1/94) of infections.

### Discussion

Rapid diagnostic tests detecting *P. falciparum* with *hrp2/3* deletion, and *P. vivax*, are a crucial tool for control where the two species are co-endemic and where deletions are frequent, such as South America and the Horn of Africa. In this

**Table 4. Sensitivity and specificity of RDTs and health center microscopy for *P. vivax* using qPCR and expert microscopy as reference. Reduced dataset of N = 524 samples with data for the First Response RDT available.**

| Diagnostic method (N = 524) | 95% LOD [CI95] | Sensitivity | | | Sensitivity vs. expert microscopy | Specificity |
|---|---|---|---|---|---|---|
| | | All densities | > 20 parasites/µL % (n/N) | >200 parasites/µL % (n/N) | | |
| Health center microscopy | 5649.9 [2723.4, 11721.3] | 35.8 [28.7, 43.2] (64/179) | 71.8 [61.0, 81.0] (61/85) | 78.2 [67.4, 86.8] (61/78) | 77.3 [66.2, 86.2] (58/75) | 98.3 [96.3, 99.4] (339/345) |
| BIOCREDIT Malaria Ag Pf/Pv (cHL/L) | 180.1 [28.1, 1152.8] | 44.7 [37.3, 52.3] (80/179) | 92.9 [85.3, 97.4] (79/85) | 98.7 [93.1, 100.0] (77/78) | 96.0 [88.8, 99.2] (72/75) | 99.4 [97.9, 99.9] (343/345) |
| Bioline Malaria Ag Pf/Pf/Pv | 113.5 [19.9, 647.9] | 45.8 [38.4, 53.4] (82/179) | 95.3 [88.4, 98.7] (81/85) | 98.7 [93.1, 100.0] (77/78) | 96.0 [88.8, 99.2] (72/75) | 98.8 [97.1, 99.7] (341/345) |
| First Response | 85.6 [14.4, 507.8] | 46.4 [38.9, 54.0] (83/179) | 96.4 [90.0, 99.3] (82/85) | 98.7 [93.1, 100.0] (77/78) | 97.3 [90.7, 99.7] (73/75) | 99.1 [97.5, 99.8] (342/345) |

study, the novel BIOCREDIT tests with a line combining HRP2 and LDH for *P. falciparum* diagnosis showed very similar sensitivity to the established Bioline RDT with separate lines for HRP2 and LDH. The combined line facilitates result interpretation, while separate lines allow the identification of infections with potential deletions that should undergo molecular typing. Both tests detected 45–48% of all *P. falciparum* and *P. vivax* infections using qPCR as gold standard, and 93–97% at densities above 20 parasites/µL by qPCR. Specificity was > 99%. All RDTs were significantly more sensitive than microscopy conducted at the health center.

The limit of detection of HRP2-based tests for *P. falciparum* was in the range of 30–60 parasites/µL, compared to approximately 250 parasites/µL for the LDH line on the Bioline RDT. The limit of detection of LDH-based tests for *P. vivax* was substantially higher, i.e., approximately 200 parasites/µL on the Bioline and BIOCREDIT RDTs.

As seen in other studies [23,24], sensitivity of health center microscopy was particularly low for mixed-species infections. RDTs performed better and detected both species in 5/12 (BIOCREDIT) and 7/12 (Bioline) infections where *P. falciparum* and *P. vivax* were present at densities >20 parasites/µL. While this is well below the > 90% sensitivity for either species across the entire dataset, the number of samples is too low to determine RDT performance for mixed-species infections.

While this is the first field test of the BIOCREDIT Ag Pf (cHL) and BIOCREDIT Ag Pf/Pv (cHL/L) RDTs, several previous studies evaluated BIOCREDIT RDTs and Bioline RDTs for *P. falciparum* and *P. vivax*, including BIOCREDIT RDTs with separate lines for HRP2 and LDH for *P. falciparum* detection. Some studies found higher RDT sensitivity compared to PCR than the current study [25–28]. However, direct comparisons of the number of PCR-positive infections that are detected by RDT among studies are not possible because different PCR assays vary widely in their limit of detection [29], and the proportion of low-density infection varies among populations. For example, BIOCREDIT RDTs with separate lines for LDH and HRP2, and a Bioline test with HRP2 only were tested in Senegal. Using PCR as reference test, sensitivity reached 62–65% for LDH, and 71–78% for HRP2 [28]. This is higher than observed in the current study, but importantly, the limit of detection of the reference PCR was approximately 10-fold higher than the one used here (3.2 vs. 0.3 parasites/µL). Thus, in the Senegal study, very-low density infections might have been missed [20, 30]. A novel RDT from Abbott with a combined HRP2/LDH line, along with the same Bioline Malaria Ag Pf/Pf/Pv test used in the current study, was recently evaluated in Ethiopia [12]. Both tests reached a sensitivity of 76–78% compared to nested PCR.

The BIOCREDIT test with separate HRP2 and LDH lines was also evaluated for *P. falciparum* diagnosis in Burundi and Ghana, using identical sample processing and qPCR protocols as in the current study. Sensitivity vs. qPCR for both lines combined was 52% in Ghana [14], and thus similar to the combined line of the test evaluated in the current study. In Burundi sensitivity vs. qPCR was 80% [13]. Likely, in Burundi the very high level of transmission, resulting in a higher pyrogenic threshold, results in overall high parasite densities in clinical patients and thus high sensitivity of the RDT [31].

In conclusion, all RDTs evaluated showed high sensitivity and specificity, but RDTs for *P. vivax* remain less sensitive than for *P. falciparum*. The sensitivity and specificity of the novel BIOCREDIT RDTs are on par with other leading RDTs. The addition of a separate line for *P. vivax* diagnosis did not affect performance of the *P. falciparum* line.

## Supporting information

**S1 File. qPCR and dPCR protocols.**
(PDF)

**S2 File. Database.**
(CSV)

**S3 File. STARD checklist for studies of diagnostic accuracy.**
(DOCX)

## Acknowledgments

The authors thank all patients who participated in this study, and the health centers personnel who supported this study.

## Author contributions

**Conceptualization:** Cristian Koepfli.

**Data curation:** Colins O. Oduma, Cristian Koepfli.

**Formal analysis:** Tolulope Adeyemi Kayode, Cristian Koepfli.

**Funding acquisition:** Cristian Koepfli.

**Investigation:** Colins O. Oduma, Birhanu Lulu, Yalemwork Ewnetu, Laurel A. Lown, Tolulope Adeyemi Kayode, Dawit Hawaria.

**Methodology:** Cristian Koepfli.

**Project administration:** Cristian Koepfli.

**Supervision:** Dawit Hawaria, Cristian Koepfli.

**Writing – original draft:** Cristian Koepfli.

**Writing – review & editing:** Colins O. Oduma, Laurel A. Lown, Dawit Hawaria.

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
