## [Decision Letter · Decision Letter 0]

23 Sep 2025

Response to Reviewers Revised Manuscript with Track Changes Manuscript

Shaden Kamhawi

co-Editor-in-Chief

Paul Brindley

co-Editor-in-Chief

**Journal Requirements:**

At this stage, the following Authors/Authors require contributions: Colins O. Oduma, Birhanu Lulu, Yalemwork Ewnetu, Laurel A. Lown, Tolulope Adeyemi Kayode, Dawit Hawaria, and Cristian Koepfli. Please ensure that the full contributions of each author are acknowledged in the "Add/Edit/Remove Authors" section of our submission form.

4) Please ensure that the funders and grant numbers match between the Financial Disclosure field and the Funding Information tab in your submission form. Note that the funders must be provided in the same order in both places as well.

**Reviewers' comments:**

**Key Review Criteria Required for Acceptance?**

**Methods**

-Are the objectives of the study clearly articulated with a clear testable hypothesis stated?

-Is the study design appropriate to address the stated objectives?

-Is the population clearly described and appropriate for the hypothesis being tested?

-Is the sample size sufficient to ensure adequate power to address the hypothesis being tested?

-Were correct statistical analysis used to support conclusions?

-Are there concerns about ethical or regulatory requirements being met?

Reviewer #1: The study objectives are clearly described and study design is satisfactory for the purpose of the study. The study population is appropriate. Sample size estimates have been done and the number of study subjects closely matches the expected sample size. Statistical analysis are reasonable.

Reviewer #2: See attached review.

Reviewer #3: 1. Insufficient methodology has been provided for the qPCR and digital PCR assays. While references are available, it would be much clearer if this manuscript included primer sequences, reaction conditions, volume, equipment and reagents used.

2. Is the impact of freezing blood on the sensitivity of the Bioline RDT known? This should be discussed as a possible impact.

3. Line 216. It is not clear how the limit of detection was determined.

**Results**

-Does the analysis presented match the analysis plan?

-Are the results clearly and completely presented?

-Are the figures (Tables, Images) of sufficient quality for clarity?

Reviewer #1: The results match analysis plan and the findings are reasonably explained well. The Tables are reasonable.

Reviewer #2: See attached review.

Reviewer #3: 4. It would be helpful to add a column in the tables that includes only samples with less than 20 parasites/ul, since it is difficult to determine without looking at supplementary files and will make it clearer to the reader that the sensitivity is very low at these densities. Alternatively, this could be described (using quantitative data) in the discussion when mentioning differences between studies and the LOD of qPCR methods used.

**Conclusions**

-Are the conclusions supported by the data presented?

-Are the limitations of analysis clearly described?

-Do the authors discuss how these data can be helpful to advance our understanding of the topic under study?

-Is public health relevance addressed?

Reviewer #1: The conclusions are consistent with study results. Appropriate discussions and public health relevance indicated.

Reviewer #2: See attached review.

Reviewer #3: 5. Line 173, it mentions 7 samples that were microscopy positive but PCR negative. Have species other than P. falciparum and P. vivax been considered and how they would impact RDT results? Is it known if they cross-react?

**Editorial and Data Presentation Modifications?**

Reviewer #1: None

Reviewer #2: See attached review.

Reviewer #3: (No Response)

**Summary and General Comments**

Reviewer #1: File attached.

Reviewer #2: See attached review.

Reviewer #3: The study by Oduma et al investigates the sensitivity and specificity of multiple antigen detection rapid diagnostic tests for the detection of P. falciparum and P. vivax malaria infections in a field setting across multiple locations in Ethiopia. Importantly, this study includes analysis of HRP2/3 and LDH sensitivity, which is of increasing importance with HRP2/3 deletion parasites spreading, as well as a significant number of subjects with very low parasitemia. The authors found differences between HRP and LDH sensitivity, as expected from previous studies, and relatively little differences in performance between the tests used. The results are clearly described and the conclusions discussed are reasonable. I have minor suggestions to increase the ease of reproducibility in terms of describing experimental methods and analyses used. There are also several points that could be discussed more clearly to help the reader easily understand the data and conclusions.

PLOS authors have the option to publish the peer review history of their article (what does this mean?). If published, this will include your full peer review and any attached files.

Reviewer #1: No

Reviewer #2: No

Reviewer #3: No

**Figure resubmission:**

**Reproducibility:** To enhance the reproducibility of your results, we recommend that authors of applicable studies deposit laboratory protocols in protocols.io, where a protocol can be assigned its own identifier (DOI) such that it can be cited independently in the future. Additionally, PLOS ONE offers an option to publish peer-reviewed clinical study protocols. Read more information on sharing protocols at https://plos.org/protocols?utm_medium=editorial-email&utm_source=authorletters&utm_campaign=protocols

---

## [Editor Report · Decision Letter 1]

7 Oct 2025

Response to Reviewers Revised Manuscript with Track Changes Manuscript

Shaden Kamhawi

co-Editor-in-Chief

Paul Brindley

co-Editor-in-Chief

**Additional Editor Comments:**

Reviewer 3.

It would be helpful to add a column in the tables that includes only samples with less than 20

parasites/ul, since it is difficult to determine without looking at supplementary files and will make

it clearer to the reader that the sensitivity is very low at these densities. Alternatively, this could

be described (using quantitative data) in the discussion when mentioning differences between

studies and the LOD of qPCR methods used.

Author Response: We are a bit puzzled by this comment. As the reviewer points out, RDTs are not

expected to detect such very-low density samples, and calculations of sensitivity at such low

density is primary influenced by the limit of detection of the PCR. Rather than adding a column to

the table, we added the following sentence to the results (line 294):

“For either species, at densities <20 parasites/µL, as expected, RDTs showed poor sensitivity of

below 10%.”

Editor comments. Please add a column in the table as suggested by the reviewer. Your response to this comment is not acceptable.

5. Line 173, it mentions 7 samples that were microscopy positive but PCR negative. Have species

other than P. falciparum and P. vivax been considered and how they would impact RDT results? Is

it known if they cross-react?

Response: We agree this is a possibility, but did not screen by qPCR for other species. Currently,

assays for P. malariae and P. ovale are not established in our PCR lab in Hawassa, thus screening

for these species is not feasible for this study.

**Reviewers' comments:**
**Figure resubmission:**

**Reproducibility:**To enhance the reproducibility of your results, we recommend that authors of applicable studies deposit laboratory protocols in protocols.io, where a protocol can be assigned its own identifier (DOI) such that it can be cited independently in the future. Additionally, PLOS ONE offers an option to publish peer-reviewed clinical study protocols. Read more information on sharing protocols at https://plos.org/protocols?utm_medium=editorial-email&utm_source=authorletters&utm_campaign=protocols

---

## [Editor Report · Decision Letter 2]

14 Oct 2025

Dear Mr. Koepfli,

We are pleased to inform you that your manuscript 'First Field Evaluation of Novel LDH- and HRP2-based Rapid Tests for Plasmodium vivax and Plasmodium falciparum Malaria Diagnosis' has been provisionally accepted for publication in PLOS Neglected Tropical Diseases.

Best regards,

Sanjai Kumar

Guest Editor

Abhay Satoskar

Section Editor

Shaden Kamhawi

co-Editor-in-Chief

Paul Brindley

co-Editor-in-Chief

---

## [Editor Report · Acceptance letter]

Dear Mr. Koepfli,

We are delighted to inform you that your manuscript, "First Field Evaluation of Novel LDH- and HRP2-based Rapid Tests for Plasmodium vivax and Plasmodium falciparum Malaria Diagnosis," has been formally accepted for publication in PLOS Neglected Tropical Diseases.

Best regards,

Shaden Kamhawi

co-Editor-in-Chief

Paul Brindley

co-Editor-in-Chief
